Short-term safety and efficacy of aspirin in patients with COVID-19: a systematic review and meta-analysis of randomized controlled trials

Xue Liwen 1
Qi Yuhan 2 3
Zou Yingying zouyingyingzyy@126.com 1 4
1 Department of Pathology and Pathophysiology, School of Basic Medicine, Kunming Medical University , Kunming , China
2 West China School of Medicine, West China Hospital, Sichuan University , Chengdu , China
3 Division of Vascular Surgery Department of General Surgery, West China Hospital, Sichuan University , Chengdu , Sichuan Province , China
4 Frontiers Medical Center, Tianfu Jincheng Laboratory , Chengdu , Sichuan Province , China
Hromić-Jahjefendić Altijana
Electronic publication date: 2025 May 21
Publication date: 2025
Volume: 13
Electronic Location ID: e19466
Received 2025 Feb 13; Accepted 2025 Apr 23
Copyright: ©2025 Xue et al.
Copyright year: 2025
Copyright holder: Xue et al.
License: This is an open access article distributed under the terms of the Creative Commons Attribution License, which permits unrestricted use, distribution, reproduction and adaptation in any medium and for any purpose provided that it is properly attributed. For attribution, the original author(s), title, publication source (PeerJ) and either DOI or URL of the article must be cited.
License URL: https://creativecommons.org/licenses/by/4.0/

Keywords: COVID-19, Aspirin, Safety and efficacy

Funding: National Natural Science Foundation of China 32060184 81760149 Yunnan Province-Kunming Medical University 202401AY070001-029 First-Class Discipline Team of Kunming Medical University 2024XKTDTS11 Scientific and Technological Innovation Team of Kunming Medical University CXTD202101 This work was supported by the National Natural Science Foundation of China (grant numbers: 32060184 and 81760149), Joint Special Funds for the Department of Science and Technology of Yunnan Province-Kunming Medical University (grant no. 202401AY070001-029), First-Class Discipline Team of Kunming Medical University (grant no. 2024XKTDTS11), the Scientific and Technological Innovation Team of Kunming Medical University (grant no. CXTD202101). There was no additional external funding received for this study. The funders had no role in study design, data collection and analysis, decision to publish, or preparation of the manuscript.

==============================
Objective

Coagulation activation and inflammatory derangements are key characteristics of coronavirus disease 2019 (COVID-19). Aspirin therapy in patients with COVID-19 remains uncertain due to conflicting evidence regarding its ability to balance anti-inflammatory and antithrombotic benefits against potential bleeding risks in the context of COVID-19-associated coagulopathy. This study aimed to compare the clinical safety and efficacy of aspirin in patients with COVID-19 in randomized controlled trials (RCTs).

Methods

In the present systematic review and meta-analysis, the Medline, Embase, and Cochrane Library databases were searched for RCTs from database inception to January 13, 2023. Data were independently extracted and screened by two authors using structured data collection forms based on published reports. Results were calculated using odds ratios (ORs) and 95% confidence intervals (CIs) with the Mantel-Haenszel method. Quality was assessed using the Cochrane Risk of Bias tool. The main outcomes were short-term all-cause mortality, bleeding events and any thrombosis events. This meta-analysis was registered on PROSPERO.

Results

A total of 922 studies were identified. Finally, six RCTs with low risk of bias were pooled in the analysis. The results showed that aspirin use was not associated with a reduction in all-cause mortality (OR = 0.95, 95% CI [0.88–1.03], I2 = 0%) or the risk of any thrombosis (RR 0.88, 95% CI [0.77–1.01], I2 = 0%), but aspirin use was associated with a higher risk of bleeding (OR 1.72, 95% CI [1.32–2.24], I2 = 0%). No obvious risk of bias was found among the included RCTs for the primary outcome.

Conclusion

Routine low-dose aspirin use does not reduce the risk of short-term mortality and risk of any thrombosis but increases the risk of bleeding. The data does not support the use of low-dose aspirin in patients with COVID-19.

Background

By of the end of 2022, there were above 663 million documented coronavirus disease 2019 (COVID-19) cases, with over 6.6 million mortality worldwide (Worldometer, 2025; Wanjari et al., 2023; Challen et al., 2021; Barber et al., 2022). Coagulation activation and inflammatory derangements are key characteristics of COVID-19 (Osuchowski et al., 2021; Leentjens et al., 2021; Maiese et al., 2021). Coagulation activation refers to the dysregulated formation of intravascular blood clots driven by endothelial injury, hyperactive platelets, and imbalanced clotting factors. This prothrombotic state predisposes patients to complications including venous thromboembolism, myocardial infarction, and ischemic stroke. Inflammatory derangements involve a maladaptive immune response characterized by excessive release of pro-inflammatory cytokines, which not only exacerbate lung injury but also create a bidirectional interplay with the coagulation system-where inflammation promotes fibrin deposition while clotting factors further amplify inflammatory cascades. It is estimated that 5–30% of COVID-19 patients have a higher risk of venous thromboembolic complications and worse progress than other acute illness, particularly myocardial infarction, and ischemic stroke (Jiménez et al., 2021; Kunutsor & Laukkanen, 2020; Klok et al., 2020; Smilowitz et al., 2021).

Up to now, several high quality randomized clinical trials (RCTs) have demonstrated the efficacy and safety of anticoagulant therapy among hospitalized COVID-19 patients (Sadeghipour et al., 2021; Ramacciotti et al., 2022; Lopes et al., 2021; REMAP-CAP Investigators, ACTIV-4a Investigators & ATTACC Investigators, 2021; ATTACC Investigators et al., 2021). Recent studies demonstrated that the use of prophylactic standard-dose anticoagulation therapy have been recommended for the prevention of thromboembolic events (Bikdeli et al., 2020). However, the latest meta-analysis of randomized controlled trials showed that escalated-dose prophylactic anticoagulation was not associated with a reduction in all-cause death and venous thromboembolism but with an increase in major bleeding (Ortega-Paz et al., 2022). Antiplatelet therapy is one way to prevent thrombosis. Aspirin, an affordable and globally available drug, may have beneficial effects in patients with COVID-19 because aspirin inhibits platelet aggregation and reduces inflammation. Indeed, a recent trial demonstrated that aspirin can reduce thrombotic events and prevent in-vitro hyperactivity in patients with SARS-CoV-2 infection (Manne et al., 2020), and aspirin was associated with lower risk of 28-day hospitalized mortality (Chow et al., 2022). However, the RECOVERY trial, the latest multi-center RCT in UK, showed aspirin cannot reduce in 28-day mortality or risk of progressing to invasive mechanical ventilation or death, but can increase the rate of being discharged alive within 28 days in hospitalized patients with COVID-19. To date, there have been several RCTs that evaluated the efficacy and safety of aspirin in treatment of COVID-19 patients, we performed a systematic review and meta-analysis of available RCTs to further determine the safety and efficacy of aspirin in patients with COVID-19.

Methods

Search strategy and selection criteria

The present systematic review and meta-analysis was conducted according to the Cochrane Handbook for Systematic Reviews of Interventions and 2020 PRISMA guidelines (Page et al., 2021). Systematic searches of published studies were conducted in the following databases from inception through to June 30, 2024 without language restrictions: MEDLINE, EMBASE, and Cochrane Library. The keywords of searching terms were listed as follows: “aspirin”, “acetylsalicylic acid”, “COVID-19”. Grey literature was not be searched. The details of search strategy were available in Appendix S1. Titles, abstracts, and full text publications were independently screened by two authors (LX and YQ), and the pre-specified inclusion criteria were as follows: (1) clinical studies reporting outcomes in patients with COVID-19; (2) we included RCTs that compared treatment with aspirin to non-aspirin in patients with or without indication for antiplatelet; (3) crossover trials were excluded from our study; (4) there was no restriction on the dosage and duration of aspirin therapy. Disagreements were resolved through discussion with a third reviewer (YZ).

Data analysis

Two independent researchers (YQ and LX), blinded to each other, collected and extracted data using a predefined extraction form for the eligible published studies, and any discrepancies were checked and resolved with a third reviewer (YZ). After screening, none of the trials were published in duplicate data.

The following baseline characteristics of included patients were collected: clinical status of patients, study design, age, sex, number of patients, and follow-up duration. The primary outcomes of interest were short-term all-cause mortality (short-term is defined as less than 60 days after intervention) and bleeding (bleeding consists of major bleeding and minor bleeding. Major bleeding is defined as fatal bleeding, symptomatic bleeding in a major organ, such as the brain, liver, kidney, spleen, lung, eye, pericardium, spine, or bleeding requiring transfusion of at least two units of blood. Minor bleeding is defined as bleeding that does not cause the patient to seek treatment, bleeding requiring a healthcare assessment, or less invasive treatment such as heavy menstrual bleeding, ecchymosis, or epistaxis). Secondary outcomes as follows: (1) any thrombosis (any thrombosis defined as pulmonary embolism, myocardial infarction, stroke, systemic arterial thromboembolism, and deep venous thrombosis; (2) deep venous thrombosis (DVT); (3) pulmonary embolism (PE); (4) stroke; (5) myocardial infarction (MI); 6) high-flow oxygen or ventilation event (patients already on high-flow oxygen or mechanically ventilated before randomization will not be counted as having had an outcome); (7) discharged alive; (8) acute kidney injury (AKI).

Based on the Cochrane Handbook for systematic review and meta-analysis, the risk of bias was independently assessed by two reviewers (YQ and LX) according to Version 2 of the Cochrane risk-of-bias tool for randomized trials (RoB 2) (Higgins et al., 2011). Disagreements were discussed and resolved by a third reviewer (YZ). To further explore the stability of the findings, sensitivity analysis will be performed by sequentially removing each single study from the pooled effect estimates. The number needed to treat (NNT) or the number needed to harm (NNH) to prevent or cause an adverse event was calculated according to the absolute risk differences.

We performed the data analyses using Review Manager Version 5.3 (The Nordic Cochrane Centre, København, Denmark) following methods described in the Cochrane Handbook for Systematic Reviews of Interventions (version 6.2). For dichotomous data, we calculated the odds ratio (OR) and 95% confidence intervals (CIs) with the Mantel–Haenszel method. Statistical heterogeneity across studies was estimated using the I2 statistic, and an I2 value less than 25% indicates low heterogeneity, I2 value between 25% and 75 indicates moderate heterogeneity, and I2 value larger than 75% indicates high heterogeneity (Higgins et al., 2003).

Results

After the literature search, a total of 922 records were identified through the electronic database search. After removing duplicates, 823 potential publications were left for further assessment. The literature selection generated 108 articles, which included six RCTs were polled in to the quantitative analysis. The PRISMA flow diagram is presented in Fig. 1 and PRISMA checklist is presented in Table S1. Six studies reported short-term all-cause mortality. Regarding the bleeding event, five RCTs reported major bleeding or minor bleeding in COVID-19 patients. The characteristics of the included studies are presented in Table 1. The results of the ROB 2 tool for RCTs showed a low risk of bias in most domains of the included studies. The results of the bias assessment are shown in Fig. S1. Disagreements were always resolved through consensus between reviewers.

Figure 1 Study selection flow diagram.

Table 1 General characteristics of the six randomized controlled trials on aspirin vs. control in COVID-19 patients.

Study, year	Clinical status	Design	Time period	Location	Sample size (treat/control)	Median age-y; aspirin/control	Male % aspirin/ control	Dose of aspirin (once daily)	Treatment time	Follow-up duration (days)	
ACTIV-4B, 2021	non-critically ill, outpatients	double-blind	2020 to 2021	52 centers in the US	164/164	54.0/54.0	42.1/41.5	81 mg	for 45 days	45	
ACT-C#, 2022	high risk, outpatients	open-label	2020 to 2022	48 centers in 11 countries	1,945/1,936	45.2/44.8	61.4/59.7	100 mg	for 28 days	45	
ACT-H*, 2022	inpatients	open-label	2020 to 2022	62 centers in 11 countries	1,063/1,056	55.0/54.8	61.1/56.1	100 mg	for 28 days	45	
RECOVERY, 2022	inpatients	open-label	2020 to 2021	177 centers in the UK	7,351/7,541	59.2/59.3	62/61	150 mg	until discharge	28	
REMAP-CAP, 2022	critically ill patients, inpatients	open-label	2020 to 2021	105 centers in 8 countries	565/529	57.0/57.0	64.8/65.4	75–100 mg	14 days	21	
REMAP-CAP-n, 2022	non-critically iIll patients, inpatients	open-label	2020 to 2021	105 centers in 8 countries	90/109	52.0/53.4	63.3/68.8)	75–100 mg	14 days	21	
RESIST, 2022	inpatients	open-label	2020 to 2021	India	221/219	53/52	72/74	75 mg	for 10 days or until discharge	28	
Notes.

# Community patients.

* Hospitalized patients.

Primary outcomes

Six RCTs were included for the primary outcome of short-term all-cause mortality, and five RCTs (comprising six cohorts) report the outcome of a bleeding event. Two cohorts reported outpatient outcomes and four cohorts reported inpatient outcomes. In outpatients, the incidence of all-cause mortality was 0.57% (12/2,089) in the aspirin group, and the incidence of all-cause mortality was 0.58% (12/2,072) in the non-aspirin group. In inpatients, the incidence of all-cause mortality was 17.17% (1,579/9,198) in the aspirin group and 17.65% (1,659/9,397) in the non-aspirin group. Overall, the incidence of all-cause mortality was 14.10% (1,591/11,287) in the aspirin group and 14.65% (1,671/11,409) in the non-aspirin group. Compared to the control group, aspirin was not associated with a reduction of all-cause mortality (OR = 0.95, 95% CI [0.88–1.03], I2 = 0%), similar results were observed in inpatients and outpatients (Fig. 2A). The incidence of bleeding event was 1.58% (149/9,428) in the aspirin and 0.92% (88/9,574) in the non-aspirin group, aspirin was associated with an increased risk of bleeding (OR = 1.72, 95% CI [1.32–2.24], I2 = 0%) (Fig. 2B). Aspirin was associated with an increased risk of bleeding in inpatients (OR = 1.74, 95% CI [1.33–2.28], I2 = 0%). Separately, aspirin was associated with a higher risk of major bleeding (OR = 1.57, 95% CI [1.19–2.07], I2 = 52%) and minor bleeding (OR = 4.51, 95% CI [1.62–12.55], I2 = 52%) (Fig. S2).

Figure 2 Forest plots of asprin treatment vs. non-asprin treatment for the all-cause mortality (A) and bleeding (B).

CI, confidence intervals; M-H, Mantel-Haenszel; IV, inverse variance.

Secondary outcomes

Based on 11,005 aspirin and 11,152 non-aspirin patients, the pooled estimate of any thrombosis rate was 3.90% (428/11,005) in the aspirin group and 4.37% (487/11,152) in non-aspirin patients. Compared to the control group, aspirin was not associated with a reduction in incidence of any thrombosis in patients with COVID-19 (OR = 0.88, 95% CI [0.77–1.01], I2 = 0%) (Fig. 3A). Further analysis revealed that compared with the non-aspirin group, aspirin group was not associated with a reduction in risk of deep venous thrombosis (DVT) (OR = 0.75, 95% CI [0.49–1.16], I2 = 0%), pulmonary embolism (OR = 0.90, 95% CI [0.78–1.05], I2 = 0%), stroke (OR = 0.85, 95% CI [0.49–1.48], I2 = 52%), or myocardial infarction (OR = 0.96, 95% CI [0.55–1.67], I2 = 0%) (Fig. S3).

Figure 3 Forest plots of asprin treatment vs. non-asprin treatment for the any thrombosis (A) and high-flow oxygen or ventilation (B).

CI, confidence intervals; M-H, Mantel-Haenszel; IV, inverse variance.

Three RCT trials, including 16,721 patients, were included for the outcomes of high-flow oxygen or ventilation event in the hospital. The results demonstrated that aspirin cannot reduce the odds ratio of high-flow oxygen or ventilation in inpatients (OR = 0.93, 95% CI [0.85–1.02], I2 = 0%) (Fig. 3B).

Three RCT trials, including 16,172 inpatients, were included to assess the outcomes of being discharged alive. The results showed that aspirin cannot reduce the odds ratio of being discharged alive (OR = 1.07, 95% CI [1.00–1.15], I2 = 0%) (Fig. S4).

Three RCT trials, including 3,881 outpatients and 16,890 inpatients , were included to assess the outcomes of acute kidney injury (AKI) event, The results elucidated that aspirin was not associated with the risk of AKI (OR = 1.01, 95% CI [0.85–1.19], I2 = 0%) (Fig. S5).

Sensitivity analyses and numbers needed to treat

Sensitivity analysis by sequentially excluding one trial at a time showed that individual study data did not significantly influence any of the included main outcomes (Table S2). The NNT and NNH calculations, reflecting the number of treated patients needed to prevent or cause each outcome, are shown in Fig. 4. The NNT for all-cause mortality was 182, for any thrombosis was 209, for PE was 280, for high-flow oxygen or ventilation was 140, for DVT was 806, for stroke was 2,580, for MI was 2,030, and for AKI was 45,642. The NNH for any bleeding, major bleeding, minor bleeding, and discharged alive events were 151, 199, and 75, respectively.

Figure 4 NNT and NNH of aspirin in patients with COVID-19.

Discussion

The present meta analysis includes all available RCTs comparing the safety and efficacy of aspirin in COVID-19. Finally, we find that the low dose aspirin: (1) is not associated with lower risk of all-cause mortality and discharged alive. (2) is not associated with lower risk of any thrombosis rate, deep venous thrombosis, pulmonary embolism, stroke, or myocardial infarction; (3) is not associated with the risk of high-flow oxygen or ventilation or acute kidney injury; (4) is associated with significantly higher risk of any, major, and minor bleeding.

Previous studies have demonstrated the significant impact of increasing the risk of hypercoagulability and thrombosis in patients with COVID-19, leading to an increase in deep vein thrombosis, pulmonary embolism, and mortality (Sivaloganathan, Ladikou & Chevassut, 2020; Chow et al., 2021; Carfora et al., 2020; Abou-Ismail et al., 2020; Katsoularis et al., 2022). Aspirin, classical medicine, is widely used for its anti-inflammatory, analgesic, antipyretic actions, and antithrombotic properties, which may be benefit patients with COVID-19. Previous several observational studies also have found aspirin had a decreased risk of mortality in hospitalized patients with COVID-19. In a previous and the largest observational cohort study, the CATAMARAN study, which rolled 112,269 US adults hospitalized with COVID-19, the results showed aspirin use was associated with lower odds of 28-day in-hospital mortality (10.2% vs. 11.8%; odds ratio [OR], 0.85; 95% CI [0.79–0.92]; P < 0.001) (Chow et al., 2022). Another study involving 12,600 propensity-matched patients from the VA health system showed outpatient aspirin prescription were associated with a significantly lower risk of 30-day mortality (OR: 0.38; 95% CI [0.33–0.45]) (Osborne et al., 2021). However, the RECOVERY trial, a RCT evaluating COVID-9 therapy, randomized 14,892 patients to either aspirin or non-aspirin in the hospital, and the results demonstrated 28-day all-cause mortality is similar in the two groups (OR: 0.96; 95% CI [0.89–1.04], p = 0.35). It cannot be neglected that participants in RECOVERT study are younger than those in the CATAMARAN study (median age 59 vs. 72 years). In addition, patients in CATAMARAN have higher rates of comorbidities and more critical illness. However, another RCT (REMAP-CAP) elucidated the efficacy of aspirin in critically hospitalized patients with COVID-19, and the final results show aspirin in critical patients cannot significantly reduce the risk of all-causes death (aspirin: 28.6% vs. non-aspirin: 32.1%). In our meta-analysis, the REMAP-CAP trial and the ACTIV-4B trial evaluated the clinical outcomes of critical patents with COVID-19, but the ACTTIV-4B trial included stable critical outpatients. Considering some heterogeneity between critical and non-critical patients, we did not perform subgroup analysis limited to number of trials including critical patients. To address the issue as thoroughly as possible, we divided the enrolled patients into inpatients and outpatients and results showed aspirin was not associated with lower risk of short-term all-cause mortality in both groups. Additionally, recent studies showed that anticoagulation therapy was not associated with the risk of all-cause mortality (Ramacciotti et al., 2022; Ortega-Paz et al., 2022). Therefore, antithrombotic therapy may not be a vital factor in reducing short-term mortality in COVID-19 patients. Furthermore, our meta-analysis of RCTs finds aspirin is not associated with lower risk of the short-term the risk of high-flow oxygen or ventilation and AKI. As mentioned above all, a low dose of aspirin may not play a role in the anti-inflammatory treatment of patients with COVID-19.

Antithrombus therapy is an important part in treatment of hypercoagulability and thrombosis. Recent studies show that anticoagulation therapy can reduced risk of the venous thromboembolism but cannot reduce the risk of PE, stroke, and MI in COVID-19 patients (Ramacciotti et al., 2022; REMAP-CAP Investigators, ACTIV-4a Investigators & ATTACC Investigators, 2021; Ortega-Paz et al., 2022). In our present analysis, the estimated incidence of the any thrombosis rate was 3.90% in the aspirin group and 4.37% in the non-aspirin group, aspirin was not associated with a reduced risk of any thrombosis. More specifically, aspirin not only was not associated with lower risk of the short-term the risk of PE, stroke, and MI, but it also was not associated with lower risk of DVT. As mentioned above, a low dose of aspirin may not play a role in antithrombotic therapy in treatment of patients with COVID-19.

More importantly, though there was a low risk of bleeding in our study, with an estimated bleeding rate of 1.58% in aspirin group and 0.92% in non-aspirin group, our present study demonstrates aspirin is associated with higher risk of bleeding, including both major bleeding and minor bleeding, similar to previous RCTs (DISCOVERY trial, REMAP-CAP trial, and ACT-H trial).

Limitation

Although this work is the first meta-analysis of RCTs concerning the safety and efficacy of aspirin use in patients with COVID-19, it is important to consider some limitations when interpreting our results. Firstly, the analyses were limited by the number of included studies and slightly different aspirin doses. However, this study included more than 22,000 participants, and each RCT was of high quality with low risk of bias, as well as included studies with aspirin doses of 75 to 150 mg, which is generally considered to be low-dose aspirin, usually with antiplatelet therapy. Secondly, we did not perform detailed subgroup analyses according to baseline characteristics. Nevertheless, primary outcome of all-cause mortality analysis is performed according to a pre-specified subgroup analysis by patients’ status (inpatients or outpatients), which may partially reflect the severity of the patient status. Finally, the present analysis only evaluated short-term outcomes because the included RCTs did not report long-term clinical outcomes. The long-term safety and efficacy of aspirin in patients with COVID-19 need to be further studied.

Conclusion

In conclusion, the present systematic review and meta-analysis showed that the low-dose aspirin was not associated with a reduced risk of short-term all-cause mortality, any thrombosis rate, being discharged alive, or requiring high-flow oxygen or ventilation. However, it is associated with a higher risk of any, major, and minor bleeding. In summary, aspirin may not play a role in reducing anti-inflammatory and antithrombosis in patients with COVID-19, and routine use of low-dose aspirin in such patients should not be considered.

Supplemental Information

Supplemental Information 1 Generated and analyzed dataset

Supplemental Information 2 PRISMA checklist

Supplemental Information 3 Sensitivity analysis of main outcomes excluding one study at a time

Supplemental Information 4 Full electronic search strategy

Supplemental Information 5 Risk of bias assessment of included trials

Supplemental Information 6 Supplemental Figures

Supplemental Information 7 This article is intended for medical practitioners, especially respiratory physicians

Additional Information and Declarations

Competing Interests

Author Contributions

Data Availability

The authors declare there are no competing interests.

Liwen Xue performed the experiments, analyzed the data, authored or reviewed drafts of the article, and approved the final draft.

Yuhan Qi performed the experiments, analyzed the data, prepared figures and/or tables, and approved the final draft.

Yingying Zou conceived and designed the experiments, prepared figures and/or tables, authored or reviewed drafts of the article, and approved the final draft.

The following information was supplied regarding data availability:

This is a systematic review/meta-analysis.

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
