# Peer review of "Short-term safety and efficacy of aspirin in patients with COVID-19: a systematic review and meta-analysis of randomized controlled trials"

_PeerJ, doi:10.7717/peerj.19466_

## Round 0.1 · original submission · Major Revisions

The authors should follow the comments provided by reviewers to improve the manuscript.

Reviewer 1 ·

Basic reporting

The author should also explain outcome of therapy from other risk factors (including disease severity and comorbidities), not only from anti thrombotic view.

Experimental design

The author adhere to PRISMA guidelines.

Validity of the findings

Are all six trials included in the study a randomized controlled trial? Are the patients randomized to receive aspirin or without aspirin; allocated in ‘aspirin group’ or ‘non-aspirin group’? How is the quality of each study? What critical appraisal tool is used to evaluate the quality of each trial?
Six trials included in the study. The author should explain the number of studies in outpatient and in inpatient setting because of disease severity differences. Aspirin’ adverse events are dose-dependent; therefore, the author should perform subgroup analysis.
Page 8, lines 75-78: inclusion criteria were as follows: 1) clinical studies reporting outcomes in patients with COVID-19; 2) We included RCTs that compared treatment with aspirin to non-aspirin in patients with or without indication for Antiplatelet; 3) Crossover trials were excluded from our study; 4) There was no restriction on the dosage and duration of aspirin therapy.

Is ‘Deep Venous Thrombosis’ a secondary outcome by itself or combine with others thrombosis? Furthermore, the author should reorder the results on page 11, lines 133-136.
Page 9, lines 91-93: Secondary outcomes as follows: 1) any thrombosis (any thrombosis defined as pulmonary embolism, myocardial infarction, stroke, systemic arterial thromboembolism, and deep venous thrombosis; 2) deep venous thrombosis (DVT);

Some of the studies didn’t have aspirin as a treatment group. How can the author use this study (the RECOVERY trial) information to draw the conclusion? Did the author get the raw data from that study?
the RECOVERY trial involving hospitalized patients with coronavirus disease 2019 (Covid-19) in the United Kingdom to assess the efficacy of different treatments using a single end point: mortality within 28 days after randomization; preliminary results are now reported in the Journal.2 A total of 11,303 patients were randomly assigned to one of four treatment groups (dexamethasone, hydroxychloroquine, lopinavir–ritonavir, or azithromycin) or to usual care. Patients could undergo further randomization to receive either no additional treatment or convalescent plasma, and those with progressive Covid-19 could be randomly assigned to receive no additional treatment or tocilizumab. (Normand, 2021)

Reference:
Normand ST. The RECOVERY Platform. N Engl J Med. 2021 Feb 25;384(8):757-758. doi: 10.1056/NEJMe2025674.

Additional comments

I do not agree with the word “cannot reduce” in page 11. Aspirin is not the main therapy in COVID-19, all outcome in this study (more) associated with disease severity and comorbidities.

Reviewer 2 ·

Basic reporting

- Authors need to confirm that all abbreviations are defined before being used for the first time. For example, COVID-19 on line 34 and RCT on line 36. Please cross-check again for other acronyms in this manuscript and give their definitions upon their first mention to avoid readers’ confusion.
- (lines 98-100) As this study was completed, please adjust the tenses used when describing the methods, in this case, it should be written in past tenses.
- Authors are suggested to proofread the manuscript after addressing all comments to avoid any typological, grammatical, and lingual mistakes and errors.

Experimental design

- (line 37) When exactly is the database inception? If it’s possible, please mention it too for more transparent data inclusion.
- (lines 105-106) “… an I2 value larger than 75% indicates high heterogeneity”, what is the indication for low and moderate I2?

Validity of the findings

- (line 113) Shouldn’t it be “RoB 2”?
- (lines 118-120) No need for a space between the number and percentage symbol, for example, 17.65%. Please revise.

Additional comments

- (lines 34-35) This sentence is hanging; please complete it by giving reasons for why the aspirin therapy is uncertain for COVID-19 patients. Also kindly pay attention to the full stop at the end of each sentence.
- Overall, there is room for improvement in the abstract. Authors should give more explanation on keywords used for related studies attainment, the statistical analysis, and the conclusion of the study.
- (line 47) I would suggest that the authors use updated data on global COVID-19 fatalities from more recent studies. So, instead of using the data from 2021, it’s much better to update it to a newly reported COVID-19 case number. For example, the authors could state that “By of the end of 2022, there were above 663 million documented COVID-19 cases, with over 6.6 million mortality worldwide.” The authors can cite these studies https://narrax.org/main/article/view/71/49 and https://pmc.ncbi.nlm.nih.gov/articles/PMC10919748/

- (line 48) Please provide the explanation of what coagulation activation and inflammatory derangements are in the introduction to provide more context to unfamiliar readers.
- (line 64) The authors stated that “…but none have systematic review and meta-analysis yet reported on the effect of aspirin therapy in COVID-19.” However, this phrase is overclaimed since I found some meta-analysis reports on the effect of aspirin in COVID-19 patients. Probably the authors should write in detail what kind of effects haven’t been explored deeper by other studies to bring the novelty of this study up to address the knowledge gap in the field.
- (lines 155-157) This sentence should be revised as it goes out from the initial objective of this study, which the authors stated that they want to see the effect of aspirin therapy on the REDUCED mortality of COVID-19 patients. the current sentence is a bit out of context as this study is clearly not about aspirin decision making in COVID-19.

---

## Round 0.2 · Minor Revisions

Please pay attention to the comments made in the section "Basic reporting" and "Additional Comments" to additionally improve your manuscript.

Reviewer 1 ·

Basic reporting

Methods
Who is TW?
Author: Liwen Xue M.D. 1*; Yuhan Qi, M.D. 2, 3*; Yingying Zou, M.D.
Page 6 lines 6-7: Disagreements were resolved through discussion with a third reviewer (YZ).
Page 6 lines 9-10: and any discrepancies were checked and resolved with a third reviewer (TW)
Page 7 lines 7-8: Disagreements were discussed and resolved by a third reviewer (TW)

Experimental design

The research question is well-defined. Some limitations were stated in the discussion section.

Validity of the findings

Conclusion are well stated.

Additional comments

Results
The author should reorganize the statement and add the incidence of bleeding events in the outpatient setting.
When we read the five sentences below: 1. The second sentence, ‘Two cohorts,’ should come out first (before the sentence, ‘In outpatients,’). 2. The total outpatients are 2089+2072; the total inpatients are 9198+9397. The incidence of bleeding events is from 9428 and 9574 patients; this total number refers to the total inpatient population.

Page 8 lines 10-17: In outpatients, the incidence of all-cause mortality was 0.57 % (12/2089) in the aspirin group, and the incidence of all-cause mortality was 0.58 % (12/2072) in the non-aspirin group. Two cohorts reported outpatient outcomes and four cohorts reported inpatient outcomes. In inpatients, the incidence of all-cause mortality was 17.17% (1579/9198) in the aspirin group and 17.65 % (1659/9397) in the non-aspirin group. Overall, the incidence of all-cause mortality was 14.10% (1591/11287) in the aspirin group and 14.65% (1671/11409) in the non-aspirin group. The incidence of bleeding event was 1.58% (149/9428) in the aspirin and 0.92% (88/9574) in the non-aspirin group.

What is the proportion of outpatient–inpatient study from the three RCT trials mentioned on page 9, lines 13 and 16?
Page 9 line 13: Three RCT trials, including 16172 patients,
Page 9 line 16: Three RCT trials, including 20771 patients,

---

## Round 0.3 · accepted · Accept

The authors submitted all required revisions and manuscript can be accepted.

Reviewer 1 ·

Basic reporting

No comment

Experimental design

The author has revised the article.

Validity of the findings

No comment

Additional comments

The author has addressed all inquiries.